# Bootstrapping and Pinning down the Root Meristem; the Auxin–PLT–ARR Network Unites Robustness and Sensitivity in Meristem Growth Control

**DOI:** 10.3390/ijms22094731

**Published:** 2021-04-29

**Authors:** Jacob P. Rutten, Kirsten H. Ten Tusscher

**Affiliations:** Theoretical Biology Group, Faculty of Science, Utrecht University, 3584 CH Utrecht, The Netherlands; j.p.rutten@uu.nl

**Keywords:** auxin, cytokinin, PLETHORA, root meristem size control, computational modeling, self-organization

## Abstract

After germination, the meristem of the embryonic plant root becomes activated, expands in size and subsequently stabilizes to support post-embryonic root growth. The plant hormones auxin and cytokinin, together with master transcription factors of the PLETHORA (PLT) family have been shown to form a regulatory network that governs the patterning of this root meristem. Still, which functional constraints contributed to shaping the dynamics and architecture of this network, has largely remained unanswered. Using a combination of modeling approaches we reveal how the interplay between auxin and PLTs enables meristem activation in response to above-threshold stimulation, while its embedding in a PIN-mediated auxin reflux loop ensures localized *PLT* transcription and thereby, a finite meristem size. We furthermore demonstrate how this constrained *PLT* transcriptional domain enables independent control of meristem size and division rates, further supporting a division of labor between auxin and PLT. We subsequently reveal how the weaker auxin antagonism of the earlier active Arabidopsis response regulator 12 (ARR12) may arise from the absence of a DELLA protein interaction domain. Our model indicates that this reduced strength is essential to prevent collapse in the early stages of meristem expansion while at later stages the enhanced strength of Arabidopsis response regulator 1 (ARR1) is required for sufficient meristem size control. Summarizing, our work indicates that functional constraints significantly contribute to shaping the auxin–cytokinin–PLT regulatory network.

## 1. Introduction

The plant hormones auxin and cytokinin, along with the PLETHORA (PLT) and Arabidopsis response regulators 1 and 12 (ARR1 and ARR12) transcription factors, all play critical roles in controlling root growth and zonation. PLT transcription factors, which form a gradient emanating from the root quiescent center, have been shown to support division and prevent differentiation [1,2,3]. A complementing domain of cytokinin signalling, mediated by ARR1 and ARR12, is centered in the Elongation Zone (EZ), where it represses division and promotes elongation and differentiation [4,5,6,7,8]. In contrast to these relatively simple transcription factor localizations auxin displays a more complex pattern, with an auxin gradient tapering off from the Stem Cell Niche (SCN) and a secondary increase of auxin occurring at the start of the elongation zone [9,10,11]. In agreement with this more complex pattern, auxin appears to play a more multifaceted role, influencing the size of the meristem while also impacting division, elongation and differentiation rates [3,12,13,14]. This suggests that besides dosage dependence, the developmental state of the cell also influences the effect of auxin.

Previously, we showed that the auxin-induced PLT transcription factors form a gradient that controls the location of the stem cell region as well as the size of the meristem in a dosage-dependent manner [3]. This PLT gradient was demonstrated to arise from a combination of high, persistent auxin signalling induced transcription centered around the QC, combined with slow, division as well as cell-to-cell movement mediated dispersal of stable PLT proteins. The slow timescale of the PLT gradient was shown to enable a division of labor, ensuring that the rapid, transient changes in auxin patterns guiding root tropic responses do not interfere with the robust PLT-dependent control of developmental zonation. In another study, it was subsequently demonstrated that the interplay of elongation zone localized cytokinin signalling with auxin transport and turnover generates an auxin minimum that coincides with the shootward boundary of the meristem, termed the transition zone boundary (TZB) [11], followed by a secondary rise in EZ. This auxin minimum spatially separates the high auxin levels in the meristem, where it controls division rate, from the elevated auxin levels in the elongation and differentiation zone, where auxin concentrations impact expansion and differentiation rates. Building on this previous work, we recently showed that the network of regulatory interactions between auxin, cytokinin signalling and PLTs is responsible for the self-organized positioning of the auxin, PLT and cytokinin signalling domains [14]. Specifically, we revealed how the mutual repression between PLT and cytokinin signalling, combined with the auxin-dependent induction of cytokinin signalling together positions the cytokinin signalling domain. Additionally, we demonstrated how by antagonizing both auxin, PLT and division, cytokinin signalling constrains the PLT domain, thereby controlling meristem size.

While the above studies addressed how meristem patterning occurs in a self-organized manner, other questions remained unaddressed. As an example, the network underlying meristem patterning involves a positive feedback loop between auxin and PLTs, with auxin inducing *PLT* transcription, while PLTs enhance auxin production and transport. While such positive feedback may be instrumental in meristem activation, how it can be simultaneously consistent with a long-term finite meristem size even in absence of cytokinin antagonisms remains an open question. Indeed, PLT transcription is in fact largely limited to the root’s SCN domain and its direct surroundings. Considering its key importance for SCN and meristem establishment and maintenance, at first sight, a larger transcription domain would seem developmentally more robust. Likely, thus far unidentified selective advantages or functional constraints have contributed to molding this highly constrained PLT transcriptional domain. On a similar note, we discerned separate auxin–PLT–ARR12 and auxin–PLT–ARR1 subnetworks involved in controlling early and late phases of meristem development [14], yet what the selective or functional advantage of such separate networks would be or which molecular mechanism underlies the functional differences between ARR1 and ARR12 remained unanswered.

In this paper, we use our previously developed multi-scale model of root developmental zonation incorporating the interplay between auxin, PLT, and ARR1/12 mediated cytokinin signalling to further unravel the regulatory logic underpinning root zonation control. We combine this with simplified Single- and Two-compartment models to further analyse auxin and PLT feedbacks, as well as a protein domain analysis of ARR1 and ARR12. Using our models we demonstrate that the positive feedback loop between auxin and PLT results in a bistable system containing both a ‘low auxin low PLT’ stable steady-state as well as a ‘high auxin high PLT’ steady state. This bistability allows for the activation of the root meristem upon germination, by enabling a suprathreshold stimulation to transfer the system from the low to the high auxin and PLT state. Additionally, this bistability ensures that after its establishment, the active ‘high auxin high PLT’ state is self-sustained and does not require persistent signalling. We next show that through pinning down the high auxin high PLT domain to the SCN, the polarized root tip auxin transport terminates the maintenance of this high auxin high PLT state as cells move away from the SCN, thus preventing unbounded meristem expansion. Instead, a finite, auxin-dependent meristem size arises, on which other factors like ARR1 and ARR12 can impinge. For these latter factors, we provide strong indications that the weaker auxin antagonism of ARR12 as compared to ARR1 arises from the absence of a DELLA interaction domain, which allows DELLA proteins to bind and co-activate ARR1. We explain the need for this initial weaker antagonism from the initial buildup phase of the meristem, when the auxin–PLT positive feedback is still transitioning from the low to the high steady-state. Once established, this high steady-state can be safely repressed more strongly without risking meristem collapse. Finally, our results further strengthen the previously demonstrated division of labor between auxin and PLT, showing that constraining PLT transcription to the SCN region enables independent control of PLT gradient and hence meristem size via the SCN and of meristem division rate via auxin.

## 2. Results

### 2.1. Introducing the Root Tip Model

Previously, we developed a multi-scale model of the Arabidopsis root tip, from here on referred to as the Root tip model. It includes known regulatory interactions between the key players—auxin, PLT, and type-B ARRs 1 and 12—as well as their downstream effects on auxin dynamics and cell growth, division, expansion and differentiation patterns. Specifically, the Root tip model includes tissue and zone dependent patterns of AUX1 auxin import, and PIN-FORMED (PIN) auxin export proteins (Figure 1B), as well as auxin dependence of AUX1 expression ((1) in Figure 1A) [15], and the effects of AUX1 ((2) in Figure 1A) [16] and PIN ((3) in Figure 1A) [17,18] on auxin levels. Together these form the auxin reflux loop that creates an auxin maximum at the SCN [3,10,19,20]. Additionally, the model incorporates the requirement that persistent, high auxin levels are required for *PLT* transcription (4 in Figure 1A) [3], and that PLT induces the expression of the auxin biosynthesis gene YUCCA3 (YUC3) ((5,6) in Figure 1A) [21,22,23,24]. Combined these generate a positive feedback loop. The PLT gradient generated by this positive feedback induces division, and represses elongation (7,8 in Figure 1A) through activation and repression of a generalized DivTF transcription factor and ElongTF transcription factor respectively [1,2,3].

Importantly, while cytokinin biosynthesis appears to occur predominantly in the root tip meristem, the downstream cytokinin signalling mediated by ARR-Bs 1 and 12 involved in controlling root tip meristem size is focused in the EZ [6,7,25,26]. In absence of sufficient data on root tip cytokinin transport, we ignored cytokinin dynamics in our model, constraining us to the expression of ARR-B 1 and 12 and their downstream effects. We model the repression of these ARR-Bs by PLT ((9) in Figure 1A), and their induction by auxin ((10) in Figure 1A) [14]. Additionally, we model the repression of PLT by ARR-12 ((12) in Figure 1A), and the repression of division, via Cyclin-dependent kinase inhibitor 2 (KRP2), by ARR-1 ((13) in Figure 1A). Finally, we incorporate the ARR-1 and ARR-12 mediated induction of the auxin degrading enzyme Gretchen Hagen 3.17 (GH3.17) ((14,15) in Figure 1A) [11,27], as well as the SHORT-HYPOCOTYL 2 (SHY2) mediated repression of polar auxin transport ((16,17) in Figure 1A) [5,6].

**Table 1 ijms-22-04731-t001:** References for Root tip model interactions.

Interaction	Refs	Interaction	Refs	Interaction	Refs
1	[15]	7	[1,2,3]	13	[14]
2	[16]	8	[1,2,3]	14	[11,27]
3	[17,18]	9	[14]	15	[11,27]
4	[3]	10	[14]	16	[5,6,7]
5	[24]	11	[28]	17	[5,6,7]
6	[21,22,23]	12	[14]	18	[29]

The thus generated model enabled us to reproduce auxin, PLT and ARR-B patterns as well as developmental meristem growth dynamics in wild type (Figure 1B, Appendix A) and mutants [14]. Intriguingly, both experimental data and our model simulations suggested that even in arr1arr12 double mutants, so in absence of significant cytokinin antagonism, meristem expansion eventually levels off.

### 2.2. A Bootstrapping Capable yet Finite Positive Feedback Loop

To investigate the mechanisms underlying this initial meristem expansion and eventual stabilization, we first tested how the Root tip model responded to various levels of shoot-derived auxin. Since we had previously established that meristem size stabilizes even without repression by cytokinin signalling, we started our investigations with *arr1arr12* double mutant settings. This enabled us to first focus on the auxin and PLT feedback dynamics. Simulations were initialized with a small, germination stage root meristem of 8 rows of cells, subjected to a variable level of shoot-derived auxin influx entering the top of the vasculature (Figure 2A,B, Appendix A). We see that below a minimum auxin influx level, PLT levels required to induce division and prevent differentiation can not be maintained and meristem collapse (differentiation of the initial cells) occurs (Figure 2C thick blue line). Beyond the auxin influx level, high PLT levels supporting divisions are built up and maintained, resulting in meristem growth (Figure 2C other thick lines).

Additionally, we observe that while a stable meristem size is reached for a wide range of auxin influx levels, the final meristem size and the rate at which this size is reached increases in a saturating manner with auxin influx level (Figure 2C bold lines, Figure A1A). After reaching a stable meristem size, this sensitivity of meristem size to auxin influx level remains, and meristem size shrinks or grows consistent with changes in auxin influx levels (Figure 2C thin lines). Importantly, the zero auxin influx level, while not supporting initial meristem activation (Figure 2C, fat dark blue line), is capable of maintaining a small stable meristem when occurring only after a certain time frame (Figure 2C, thin blue lines). Similar results were obtained when varying the level of YUCCA-mediated auxin production instead of shoot-derived auxin influx (Figure A1B). Thus, we observe a limited spreading of the high-auxin–high PLT meristematic domain which size can be tuned both by changes in local production and distal influx of auxin.

The observed threshold and hysteresis behaviour is reminiscent of the bistability present in non-linear positive feedback systems. Indeed our model incorporates non-linear positive feedback between the auxin hormone and the PLT transcription factors, in which auxin induces PLT transcription [3] and PLTs enhance auxin levels through inducing auxin synthesizing and transporter proteins and repressing auxin degrading proteins [24].

Incorporating these auxin PLT interactions in a simplified, Single compartment ODE model (Appendix A) enabled us to analytically probe the consequences of these interactions. Phase plane analysis of the simplified model shows that at low auxin influx the system is bistable, and that increasing auxin influx eventually leads to a bifurcation, after which, only the high PLT–Auxin steady state exists (Figure 2D,E). From this, it follows that upon germination, auxin transported from the cotyledon [30], incorporated in our model as an increase in auxin influx, causes the root system to transition from the low auxin–low PLT state to the high auxin–high PLT state that supports division onset. Even when auxin influx levels subsequently drop, this high auxin–high PLT state will be maintained, as is the case in planta [31,32]. In other words, given sufficient initial auxin availability, the auxin–PLT positive feedback enables a bootstrapping of the meristem, resulting in meristem activation and expansion.

### 2.3. Pinning down the High Auxin–High PLT Domain

The Single-compartment model was thus able to explain the observed bistability and hysteresis of the full Root tip model. However, in the Root tip model, we observed the occurrence of finite, tunable meristem sizes. This contrasts with the theoretically predicted infinite meristem expansion that would occur in simple, spatially extended growing tissue as predicted by our Single-compartment model. In absence of resource limitation (each new cell can produce auxin and express PLT), bistability ensures the inheritance of the high auxin high PLT state after each division event (Figure 2F,G). As a consequence, after division both daughter cells retain division competence, causing unlimited, exponential meristem size expansion. The fact that we instead observe saturating meristem growth dynamics in the Root tip model points to the presence of additional factors controlling meristem growth.

We first reasoned that actual meristem expansion in planta and in the Root tip model could, at least in part, be constrained by growth-induced PLT dilution. PLT dynamics are governed by slow, high auxin levels requiring expression combined with slow turnover. As a consequence, upon cell growth and division, cytoplasmic and nuclear volume increases cause PLT levels to become diluted. For cells that are displaced through growth to lie outside the high auxin-dependent PLT expression domain centered on the QC, these diluted PLT levels do not become automatically restored through high PLT transcription. Instead, in these cells, the diluted PLT levels result in limited induction of auxin production which in turn only slowly and to a limited extent induces PLT expression. This causes a net decrease of PLT levels after each new round of division, thereby limiting division-mediated growth of the high PLT domain. To investigate this hypothesis we removed division-induced PLT dilution from our model, something that is impossible in planta, observing substantially enhanced, but still limited and auxin sensitive meristem size expansion (Figure 3A–C, Appendix A). Strikingly, we observe that while the PLT protein domain was able to expand substantially more in absence of dilution, the high auxin domain and *PLT* transcription domain remained largely constrained to the SCN area. Thus we can exclude PLT dilution as the main determinant of limited meristem expansion in the Root tip model.

Based on our observation that auxin localization restrained the PLT transcription domain even in absence of PLT dilution, we reasoned that the PIN mediated auxin reflux loop, which focuses the largest fraction of root tip auxin around the quiescent center (QC), might be the necessary factor to limiting meristem expansion. We tested this hypothesis by reducing the strong basal polarity of vascular PINS normally occurring in planta. Specifically, we replaced all PINs across all cell files with a universal but reduced apolar PIN (to keep total cell level export equal). Next, we re-introduced a weak polarity by increasing the basal PIN level in the vasculature by 10% (to ensure the high auxin domain starts at the tip end of the meristem). In these minimally polar PIN settings, we now observe fully bistable meristem growth dynamics, with, depending on parameter settings, either meristem collapse (Figure 3D,E) or unbounded meristem growth (Figure 3F,G, Appendix A). These results indicate that in absence of the highly focused auxin patterning typical of the root tip reflux loop, the positive auxin–PLT feedback causes the high auxin–high PLT domain to keep expanding, even with PLT induction delays and division induced dilution.

We thus show the polar transport to be necessary to pin down the auxin PLT positive feedback loop. To also show that it is sufficient, we extend the simplified, Single compartment ODE model to two connected cell compartments, one representing cells in theSCN and one representing cells in the transit-amplifying (TA) domain (Figure 4A). The two compartments are connected via auxin transport, for which we varied the extent of polarity. Consistent with the observed bistability in Figure 3D–G, in apolar, steady-state conditions, depending on auxin influx level both compartments converge on either a low auxin low PLT (Figure A2A) or a high auxin high PLT state(Figure A2B). The slightly lower auxin and PLT levels occurring for the high steady-state in the TA compartment result from the enhanced PLT dilution induced by its higher division rate relative to the SCN (Figure A2B). If we next explicitly model an SCN division event, producing one daughter cell which remains in the SCN and one which enters the TA domain, we can see that the new TA cell converges to the TA specific high steady state (Figure 4B). Thus, in an a meristem with low, apolar auxin transport, the division of a high auxin high PLT cell leads to inheritance and maintenance of this high auxin high PLT state in daughter cells (high begets high) irrespective of their new position. This corresponds to the theoretically predicted unbounded meristem expansion for simple, spatially unstructured bistable systems.

Raising the polar transport rate resulted in Figure 4C, where after an SCN division, the new TA cell experiences a rapid drop in auxin levels. This is due to its leaving of the SCN domain where most auxin is now directed towards. As a result PLT levels gradually decline. A bifurcation analysis of the Two-compartment model along the parameter governing the polarity of auxin transport shows that at sufficiently high polarity, the high PLT equilibrium disappears for TA cells and only the low PLT state remains (Figure 4D). Note that similar bifurcation dynamics occur in a Single-compartment model in which we incorporate, as a proxy for an increasing auxin transport polarity level, an increasing auxin loss term (Figure A2C). We can then use a 2D phase plane to visualize the dynamics of a single cell that after division leaves the SCN and enters the TA as going from one auxin nullcline to another (Figure 4E), demonstrating the loss of the high auxin–high PLT steady state. Finally, increasing the level of shoot auxin influx increases the time it takes for the TA compartment PLT levels to drop down to the meristematic division supporting threshold (Figure 4F and Figure A2D,E). Due to ongoing cell divisions, this time translates into a distance from the QC and hence meristem size. Taken together, our analysis explains how under polar auxin transport, the funneling of auxin towards the SCN causes the more shootward cells to eventually transition to a low auxin–low PLT state causing them to lose their meristematic status and enter the EZ, thereby putting a halt to meristem expansion. Additionally, it explains how, through affecting the timing of PLT decline, this final meristem size remains sensitive to auxin influx level.

### 2.4. ARR1–ARR12 Division of Labor

Having established that the expansion of the meristem in the Root tip model is limited through strongly polarized auxin transport, we next turn to how this finite meristem size is further controlled by cytokinin signalling through ARR 1 and ARR12.

Our previous study revealed that in order for our model to reproduce experimentally measured meristem size dynamics, we had to assume that ARR1 not only activates later but also more strongly antagonizes auxin levels than ARR12 [3]. Thus, an important question is what causes these differences in auxin antagonism strength. Importantly, there is no indication that this difference is due to a difference in the DNA binding domains (Figure 5A).

The delayed activity of ARR1 arises from its DELLA-dependent transcription, with DELLA becoming derepressed only at 5dpg when the levels of its repressor giberellin have sufficiently decreased [6]. Interestingly, in addition to inducing ARR1 expression, DELLAs also interact with ARR1 thereby enhancing its effectiveness as a transcription factor. Similar interactions with DELLA have been reported for ARR2 and ARR14 [33]. We hypothesized that if ARR12 is not only independent of DELLA for its transcription, but also does not interact with DELLA for its downstream effectiveness as a transcription factor this could explain its less strong auxin antagonism.

To investigate this we set out to locate and identify the binding motif for DELLA on ARR1. All members of the type-B ARR family (to which ARR1, ARR10 and ARR12 all belong) previously shown to interact with DELLA do so with the tail end of the protein [33]. We performed a MEME motif search [34] on the various Arabidopsis type-B ARRs to see if we could find a motif in this region. Interestingly, while both ARR12 and ARR10 have conserved domains within this region, ARR1 (and all other type-B ARRs known to bind DELLA) lack these domains (Figure 5B, Figure A3). Together, this strongly supports a distinction between ARR12 as early acting, non-DELLA enhanced type-B ARR and ARR1 as a late acting, DELLA-induced and DELLA-enhanced type-B ARR.

With the molecular basis for the difference between ARR1 and ARR12 auxin antagonism strength established, we set out to investigate the functional significance of having type-B ARRs that not only differ in timing but also affect strength. To investigate this we first quantified the effects of varying ARR12 and ARR1 strengths over a 4-fold range, while keeping their timing dynamics as in the original model (Figure 5C). As expected, we find that compared to wild type parameter settings, changing ARR12 strength has an effect on meristem size from early stages onwards, while changing ARR1 strength only has an effect in later stages. Additionally, we see that the ARR1 effect on meristem size is approximately two-fold larger than that of ARR12, consistent with experimental findings as well as our earlier results [4,6,14].

Next, we ran a series of alternative model simulations. In one hypothetical scenario, we let ARR1 which normally becomes active from 5dpg activate already at 36hpg. In another scenario, we let ARR12 which normally activates at 36hpg become active only at 5dpg. That is, in these scenarios we gave the ARRs the timing properties of one another. We compared the outcomes to those of simulations with normal ARR12 and ARR1 timing dynamics. For the wild type and early acting ARR1, simulations were run for two conditions, differing in the amount of shoot-derived auxin influx (Figure 5D). We see that a later acting ARR12 eventually converges to wild type meristem size, yet initially overshoots the final steady state meristem size. In contrast, if ARR1 becomes active early in root meristem development, for high auxin influx a substantially slower increase towards final meristem size occurs, whereas for lower auxin influx a meristem collapse can be observed.

### 2.5. Impact of PLT Mobility on Meristem Size Depends on Location

Above, we established how the auxin–PLT feedback loop enables meristem activation, while its interaction with the PIN reflux loop constrains the PLT transcriptional domain to the SCN thereby limiting meristem expansion. A remaining open question is what the functional consequences of two separate PLT domains, the SCN serving as a PLT producing “source” and the TA “sink” merely inheriting and receiving PLT protein, are.

In our earlier work, using a simplified, rectangular root topology and without cytokinin signalling we observed that meristem size increases as a function of higher PLT cell-to-cell movement rates [3]. In our current model, we obtained similar results (Figure A4A–D). Next, we investigated the consequences of increasing plasmodesmatal aperture, thereby enhancing PLT mobility, either only in the SCN (Figure 6A blue lines) or only in the TA domain (Figure 6A red lines) or both (Figure 6A pink-purple lines).

Interestingly, when only increasing cell-to-cell PLT movement in the stem cell region, we observe a larger increase in meristem size and extension of the PLT gradient than when we increased PLT mobility in both the SCN and in the TA (Figure 6A–C, Figure A4E–H,M). In contrast, increasing PLT cell-to-cell movement only in the TA reduces meristem size, while reducing TA PLT mobility increases meristem size (Figure 6B, Figure A4E,I,L). This opposite effect explains why when increasing PLT mobility across the entire root, a smaller meristem size increase occurs then when this PLT mobility increase is limited to the SCN.

### 2.6. Effect of Division Rate on Meristem Cell Production Depends on Location

Root growth rate ultimately depends on the size of the meristem, which sets the number of dividing cells, and the rate of division of these meristem cells. Together these determine the cellular output rate of the meristem. In this context, the division rate of rapid transit-amplifying divisions is seen as a key determinant of meristematic output, while the importance of the rate of stem cell divisions is rarely considered. Based on the observed importance of the SCN domain as a PLT source, with PLT mobility in the SCN having a strong impact on PLT gradient and meristem size, we decided to reconsider these assumptions.

We see that increasing SCN division rate by lowering cell cycle duration from 20 h to 14 h increases the number of transit-amplifying meristem cells (Figure 7A,B, Appendix A) and thereby meristematic cell production rate (Figure 6C, while increasing cell cycle duration to 32 h decreases meristem size and production (Figure 7A–C). In contrast, meristem size is considerably less affected by similar-sized changes in TA division rates (cell cycle duration varied to 6 and 16 h compared to the standard 10h) (Figure 7D,E, Appendix A), while meristem production rate does increase significantly with TA division rate (Figure 7F). Importantly, while TA division rates impact meristem output directly through the altered division rate of the majority of meristemic cells, the impact of SCN division rate on meristem output via gradually affecting meristem size acts more slowly (Figure 7A,C,D,F).

## 3. Discussion

### 3.1. Background

The importance of the plant hormones auxin and cytokinin, as well as the PLT transcription factors in controlling root developmental zonation as well as root growth rate, has long been appreciated [1,2,3,4,5,35]. Still, how they operate together in a coordinated, self-organized manner, has thus far remained unclear.

In a recent study, we demonstrated that the auxin dependence of ARR1 and ARR12, combined with their reciprocal antagonism with auxin and PLTs, play a key role in patterning the meristematic PLT and EZ cytokinin signalling domain [14]. In the current study, we investigated which selective pressures and functional demands may have shaped the architecture and dynamics of this Auxin–PLT–ARR network. To achieve this we perturbed key aspects of network architecture and dynamic as well as key characteristics of root tip development relative to the normal in planta conditions and characterized their functional consequences. Additionally, we combined simulations of our detailed multi-scale Root tip model with simplified Single and Two-compartment models capturing the essence of auxin and PLT dynamics that are more amenable to mathematical analysis.

### 3.2. A Pinned down Positive Feedback Loop

Through combining our Root tip model with a simplified Single-compartment model we demonstrated that the positive feedback between auxin and PLT can give rise to bistability. This bistability enables the switching of the root meristem from an inactive low auxin–low PLT state, to an active, division supporting high auxin–high PLT state when stimulated beyond a certain threshold, as occurs during seed germination.

Still, despite dividing meristem cells inheriting the maternal high auxin–high PLT state, both experimental data and our Root tip model simulations indicate that even in absence of antagonistic cytokinin signalling, after an initial growth phase meristem size stabilizes. Expanding our Single compartment to a Two-compartment model incorporating auxin transport we subsequently demonstrated that this stabilization of meristem size arises from the PIN mediated focusing of auxin around the QC. Due to this focusing of auxin, the division induced displacement of cells away from the SCN results in a substantial drop of auxin levels and a subsequent slower decrease in PLT levels that prevent the unlimited maintenance of the initially inherited high auxin–high PLT state

Thus, locally, for an initial small meristem, the feedback between auxin and PLT enables an all-or-none bistable switch, while globally, for a larger meristem the presence of the auxin reflux loop prevents limitless meristem expansion and instead allows for a graded, auxin-dependent finite meristem size. This combination of all-or-nothing and graded behavior, with the all-or-nothing behavior relying on positive feedback and the gradedness on the spatially or temporally graded nature of recruiting localized positive feedback, has been observed in multiple biological systems. Examples are the calcium-induced calcium release in cardiac muscle cells which amplitude governs muscle contraction strength [36,37] and the plant vernalization pathway that enables plants to determine whether a prolonged period of cold has passed and subsequent increases in temperature are hence indicative of spring [38]. In calcium-induced calcium release, the opening of individual endoplasmic reticulum calcium channels displays all-or-nothing behavior in response to local calcium levels while the number of channels recruited to open depends in a graded fashion on the total cellular calcium current [36,37]. Similarly, in vernalization, individual cells switch their epigenetic state to memorize their exposure to cold in an all-or-nothing non-reversible manner, yet since this switching is probabilistic the number of cells depends in a graded manner on the duration of the cold period [38].

### 3.3. A Bi-Partite Meristem Growth Brake

Mutual repression is a frequently encountered motif in developmental patterning systems in which a boundary between two alternative cell fates or behaviors is being imposed, with the Drosophila segmental patterning systems being one of the most well-studied examples [39,40,41,42,43]. In our previous study, we found that ARR12 antagonizes PLTs through indirectly repressing their transcription, as well as antagonizing auxin. In addition, we established that, to explain both wild type and mutant phenotypes, in addition to antagonizing cell division via KRP2, ARR1 must have a stronger antagonistic effect on auxin dynamics than ARR12 [14]. Here, we demonstrated that despite their high sequence similarity, ARR1 and ARR12 significantly differ in the region ARR1 uses to bind to DELLA. Given the importance of DELLA in enhancing downstream ARR-B effectiveness, the absence of ARR12 DELLA binding explains the weaker effect of ARR12 over ARR1 on their shared downstream targets SHY2 and GH3.17.

Combined, these ARRs provide for the regulation of root tip meristem size, in addition to the reflux loop-mediated constraint. Importantly, ARR12 and ARR1 do not merely differ in auxin antagonism strength but also the time of action, with ARR12 becoming active from 36 hpg, while ARR1 is active only from 5dpg. Here, we used our model to demonstrate that in absence of normal, early ARR12 activity, meristem size initially overshoots final stable meristem size, whereas non-normal, early ARR1 activity causes the meristem to become vulnerable to collapse. This indicates that early in development when the auxin PLT feedback loop is gearing up ARR activity is needed to control meristem growth yet can not antagonize auxin too strongly. Later, when the high auxin high PLT state is firmly established, a stronger antagonism is needed to sufficiently control meristem size.

### 3.4. The SCN as A PLT Control Center

Previously, we established the importance of a limited, prolonged high-auxin dependent PLT transcriptional domain combined with a longer range, division and cell-to-cell motion dependent PLT protein gradient [3]. We demonstrated that this enabled a separation of timescales, in which auxin can directly and rapidly affect root growth direction, while only slowly affecting root zonation dynamics via the PLTs.

Here we further probed the relevance of the PLT transcribing SCN and non-transcribing proximal meristem subdivision by independently altering the division rate or plasmodesmatal aperture in either of these two domains. Model outcomes show that the SCN is in control of the precise translation of auxin signals into a PLT gradient. Through affecting SCN PLT transcription level, plasmodesmatal aperture or division speed, meristem size is affected, while changes in division or PLT transport in the rest of the meristem hardly affect meristem size. This enhances the previously established division of labor between auxin and PLTs [3], showing that PLT patterning is not only robust against transient changes in auxin patterning mediating, e.g., tropic responses but also to changes in division rate and cell–cell connectivity (potentially downstream of auxin) in the rest of the meristem. This further contributes to our understanding of how auxin can simultaneously control a multitude of processes, such as meristem size (via PLT), tropisms and meristematic division rate.

A potential limitation of our model is that it does not incorporate plasmodesmatal auxin transport [44], and active apolar export [45,46], causing it to slightly overestimate reflux-loop induced concentrating of auxin. This is counterbalanced by the fact that our model also ignores the WOX5 dependence of PLT transcription [47,48], which would have further constrained PLT transcription to the SCN. We are thus confident that our model output closely matches in planta PLT transcription.

Interestingly, under phosphate starvation, enhanced uptake of iron results in reduced plasmodesmatal aperture and subsequent reduction of main root meristem size [49]. Thus far, this has largely been attributed to reduced motility of SHORT-ROOT near the QC, yet based on the results presented here, decreased PLT movement may likely contribute to this phenotype. On a similar note, regulation of plasmodesmatal aperture has been shown to play a critical role in lateral root development, with enhanced callose deposition resulting in an increase of nearby spaced lateral roots [50]. Interestingly, similar phenotypes have been observed in plt3,5,7 mutants [51]. Our findings here suggest that it is the reduced motility of PLT protein rather than a downstream signal that explains the reduced inhibition of nearby lateral root formation.

Summarizing, we uncovered how the auxin-PLT positive feedback loop embedded in the root tip auxin reflux loop enables the robust and reproducible activation of finite meristems, with additional size control by ARR12 and ARR1 tuned to early and late phases of meristem development. Additionally, we discovered a hitherto underappreciated role for the SCN in controlling meristem size and output through variations in local division rates and plasmodesmatal aperture, enabled through the confinement of PLT transcription to this niche.

## 4. Materials and Methods

In this paper, we use 3 distinct but related models to better understand how the feedbacks between Auxin, PLT and the ARR-Bs control root tip growth dynamics.

### 4.1. Root Tip Model

The first model is a 2D anatomically detailed Root tip model based on the Salvi et al. 2020 model, enhanced and altered to better fit experimental observations. In this model, cells can grow and divide into daughter cells, undergo vacuolar expansion and differentiation. Cells make these developmental decisions based on the state of their hormonal-gene regulatory network. For a more detailed overview of the model and its parameters see the Appendix A. Auxin influx parameters for Figure 2 are found in Appendix A. The alternative Root tip model settigns used for Figure 3 can be found in Appendix A. The altered ARR parameters used in Figure 5 can be found in Appendix A. The diffusion parameters and plasmodesmatal patterns used in Figure 6 are found in Appendix A respectively. The altered division rates for the SCN and TA used in Figure 7 can be found in Appendix A and an overview is given in Appendix A.

### 4.2. Single-Compartment Model

The second model is a single compartment ODE system that captures in a simplified manner the essence of the Auxin–PLT positive feedback loop dynamics The model on the one hand serves as a minimal model enabling detailed phase plane and bifurcation analysis, while on the other hand allows us to identify which properties of the multi-scale Root tip model are of a more complex, emergent nature. The Single-compartment model is described by the following set of equations:(1)$$\frac{dAuxin}{dt}=\frac{p_{Aux}*Plts^{2}}{Plts^{2}+KM_{Plts,Auxin}^{2}}-d_{Aux}*Auxin+influx$$
(2)$$\frac{dPlts}{dt}=\frac{p_{Plts}*Auxin^{3}}{Auxin^{3}+KM_{Auxin,Plts}^{3}}-d_{Plts}*Plts$$
where *p_Aux_* is the auxin production rate, *KM_Plts,Aux_* is the *PLT* concentration for which auxin production is half maximum, *d_Aux_* is the auxin degradation rate, influx is the auxin influx rate, *p_Plts_* is the *PLT* production rate, *KM_Auxin,Plts_* is the Auxin concentration for which *PLT* production is half maximum, and *d_Plts_* is the *PLT* degradation rate. For the derivation of these equations see the Appendix A. The default parameters used in this paper are given in Table 2 and Appendix A. Alternative parameters settings can be found in Appendix A.

### 4.3. Two-Compartment Model

The Two-compartment model is an extension of the above described Single-compartment model, generated through coupling two compartments to model an SCN compartment and a TA compartment connected through auxin transport. To model the effect of cell growth and division, we incorporate a PLT protein level division rate-dependent dilution term. In addition, we add PIN-mediated transport between the two compartments of which the polarity/directionality can be varied, and shoot-derived auxin influx into TA compartment. Combined this results in the following system of equations:(3)$$\frac{dAuxin_{SCN}}{dt}=\frac{p_{Aux}*Plts_{SCN}^{2}}{Plts_{SCN}^{2}+KM_{Plts,Auxin}^{2}}-(d_{Aux}+Trans_{apolar})\;*Auxin_{SCN}+\left(Trans_{apolar}+Trans_{polar}\right)*\;Auxin_{TA}$$
(4)$$\frac{dAuxin_{TA}}{dt}=\frac{p_{Aux}*Plts_{TA}^{2}}{Plts_{TA}^{2}+KM_{Plts,Auxin}^{2}}-(d_{Aux}+2*\;Trans_{apolar}+Trans_{polar})\;*Auxin_{TA}+Trans_{apolar}*\;Auxin_{SCN}+influx$$
(5)$$\frac{dPlts_{SCN}}{dt}=\frac{p_{Plts}*Auxin_{SCN}^{3}}{Auxin_{SCN}^{3}+KM_{Auxin,Plts}^{3}}-(d_{Plts}+dil_{SCN})*Plts_{SCN}$$
(6)$$\frac{dPlts_{TA}}{dt}=\frac{p_{Plts}*Auxin_{TA}^{3}}{Auxin_{TA}^{3}+KM_{Auxin,Plts}^{3}}-(d_{Plts}+dil_{TA})*Plts_{TA}$$
where *Trans_Apolar_* is the baseline apolar auxin transport rate between cell compartments and *Trans_Polar_* is the rate at which auxin is rapidly transported from the *TA* to the *SCN*, *dil_SCN_* is the effective growth dilution rate in the *SCN* (assuming a 22 h division rate) and *dil_TA_* is the effective growth dilution rate in the *TA* (assuming an 11 h dilution rate). The default parameters of the Two-compartment model used in this paper are given in Table 3. A full parameter overview can be found in Appendix A. Alternative auxin influx parameters used inf Figure 4F can be found in Appendix A. 

For the Single and Two-compartment models, we used GRIND (https://tbb.bio.uu.nl/rdb/grindC.html, accessed on 12 March 2021) developed by Prof. Rob de Boer (Utrecht University, Utrecht, The Netherlands) [52] to run the ODE system, analyze phase planes and draw bifurcation diagrams.

### 4.4. MEME Analysis

We aimed to compare the known DELLA binding ARR-Bs ARR1, ARR2, and ARR14 to the two ARR-Bs known to be present and active earlier in development when Gibberellin levels are still high and therefore DELLA levels are low [6,26], namely ARR10 and ARR12. For this, we collected the Arabidopsis thaliana amino acid sequence of each of these ARR-B proteins from the UniProt database. We next used these sequences to search for related sequences in other angiosperm species using a BLAST-based sequence comparison. We used the thus obtained set of sequences to search for motifs, using a MEME analysis with the options of Table 4. To ensure that we were not missing motifs, we reran the MEME analysis forcing it to search the truncated ARR sequences for 5 domains. This merely resulted in the splitting up of the earlier found domains into subdomains rather than the discovery of new domains (see Figure A3).

## Figures and Tables

**Figure 1 ijms-22-04731-f001:**
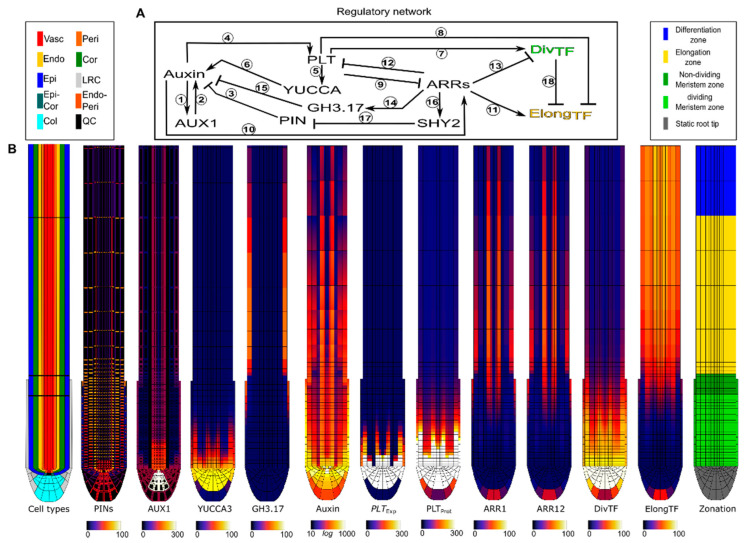
(**A**) Regulatory network present in each cell in the Root tip model. Experimental data supporting the interaction can be found in Table 1. (**B**) Spatial profile of the factors in the Root tip model at 6dpg, where meristem size has stabilized, under wild type conditions (See Appendix A for the establishment of this stable state).

**Figure 2 ijms-22-04731-f002:**
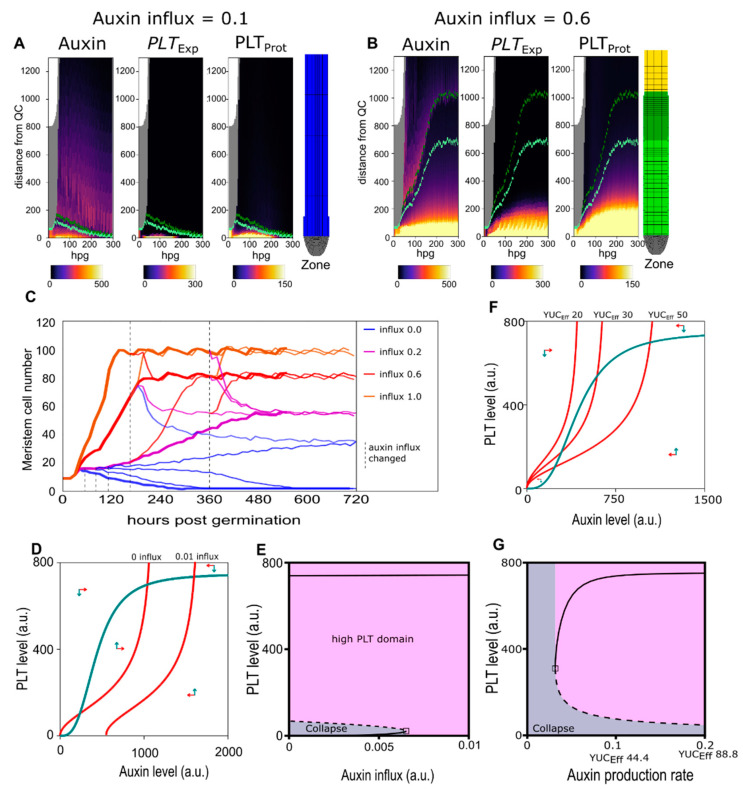
(**A**) Auxin (left panel), PLETHORA (PLT) expression (middle panel), and PLT protein (right panel) levels in the epidermis over time (in hours post-germination) for a run with 0.1 influx. Final root layout shown on the right. (**B**) Auxin (left panel), PLT expression (middle panel), and PLT protein (right panel) levels in the epidermis over time for a run with 0.6 influx (the default level) Final root layout shown on the right. (See Appendix A a more detailed timeseries of (**A**) and (**B**). (**C**) Meristem cell size trajectories for various shoot influx levels (thick lines). At several intervals (dotted lines) the influx levels were altered (thin lines). (**D**) Phase plane with nullclines for the Single-compartment model of Equations 1 and 2 with PLT nullcline in green and two auxin nullclines in red, one where shoot auxin influx is 0, and one where this influx is 9% of the roots maximum auxin production rate (0.01). Arrows indicate general direction of trajectories in that region of the phase plane. (**E**) Bifurcation diagram of the Single-compartment model for PLT level as a function of shoot auxin influx. Stable steady states depicted as continuous lines and instable steady state as a dashed line. Initial conditions converging to the high steady state are colored pink, and those converging to the lower stable steady state are colored green. (**F**) Phase plane with nullclines for the Single-compartment model of Equations 1 and 2 with PLT nullcline in green and three auxin nullclines in red, one where YUCCA-mediated auxin production is 20 times, one where it is 30 times and one where it is 50 times the basal auxin production rate. Arrows indicate general direction of trajectories in that region of the phase plane. (**G**) Bifurcation diagram of the Single-compartment model for PLT as a function of YUCCA-mediated auxin production levels.

**Figure 3 ijms-22-04731-f003:**
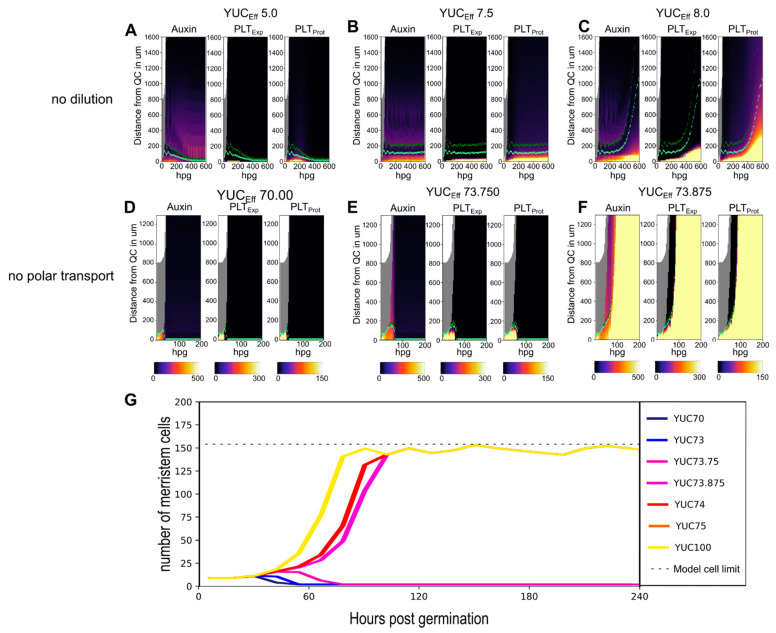
(**A**,**C**) Root tip model simulations without division-driven dilution of PLT. Auxin (left panel), PLT expression (middle panel), and PLT protein levels (right panel) for a simulation with a YUC_Eff_ of 5 (**A**), a YUC_Eff_ of 7.5 (**B**), and a YUC_Eff_ of 8 (**C**). (**D**–**F**) Root tip model simulations without polar transport. Auxin (left panel), PLT expression (middle panel), and PLT protein levels (right panel) for a simulation and a YUC_Eff_ of 70 (**D**) without polar transport and a YUC_Eff_ of 73.75 (**E**) without polar transport and a YUC_Eff_ of 73.875 (**F**). (**G**) Meristem cell number trajectories for various simulations without polar auxin transport at various YUC_Eff_ levels. The dotted line indicates the maximum number of meristem cells that can exist in the simulation window. See Appendix A for a more detailed evolution of the model over time for the no dilution (**B**,**C**) and no polar transport (**E**,**F**) case respectively.

**Figure 4 ijms-22-04731-f004:**
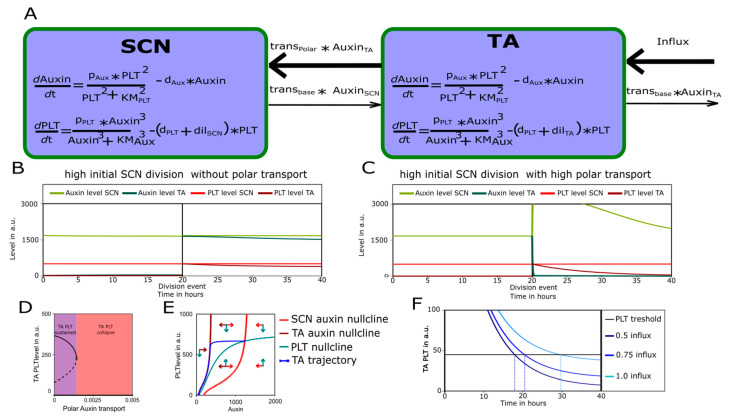
(**A**) Schematic representation of Two-compartment model with an Stem Cell Niche (SCN) and a transit-amplifying (TA) compartment. (**B**,**C**) Simulated SCN division after 20 h replacing the TA compartment in a Two-compartment model with low (**B**) or high (**C**) polar transport. (**D**) Bifurcation diagram of TA compartment PLT levels for various polar auxin transport levels. Continuous lines show stable steady state and dashed line shows instable steady state. Purple indicates polar auxin transport values for which the TA compartment can maintain a high PLT state (as in **B**). Red indicates polar auxin transport values for which the TA cannot maintain a high PLT state (as in **C**). (**E**) Phase plane with nullclines for the single-compartment model with two auxin nullclines. The PLT nullcline and trajectories are shown in green, the SCN auxin nullcline and trajectories are shown in red, and the TA auxin nullcline and trajectories, where the effect of polar auxin transport is modelled as an additional loss of auxin, are shown in dark red. (**F**) Post division TA compartment PLT dynamics for various shoot auxin influx levels (in dark to light blue). The horizontal line shows the PLT threshold that is reached at different timepoints after that TA cell has left the SCN (dashed lines).

**Figure 5 ijms-22-04731-f005:**
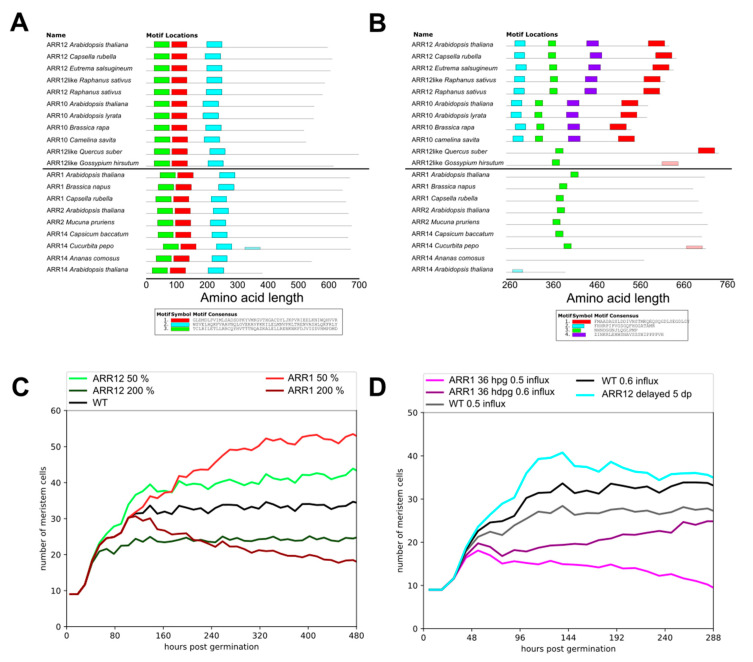
(**A**) Output of MEME analysis of the full known type B ARRs that bind co-activator DELLA (ARR1,ARR2,ARR14) in Arabidopsis and related species as well as ARR10 and ARR12, which are known to be active in the root prior to significant DELLA presence. Three domains are found on the N-terminal half of the sequence, while DELLA is known to bind the C-terminal half of the sequence. (**B**) MEME analysis of sequences from A where the first 260 amino acids were excised. ARR12 and ARR10 have two conserved domains that ARR1, ARR2, and ARR14 lack in the region where DELLA is known to binds. (**C**) Meristem cell number trajectories for various ARR12 and ARR1 expression levels compared to WT. (**D**) Meristem cell number trajectories for simulations where either ARR12 is expressed form 36 hpg and ARR1 is expressed form 5 dpg (black and gray) ARR12 is delayed to 5dpg along with ARR1 (lightblue), ARR1 is expressed from 36 hpg along with ARR12 (purple and pink).

**Figure 6 ijms-22-04731-f006:**
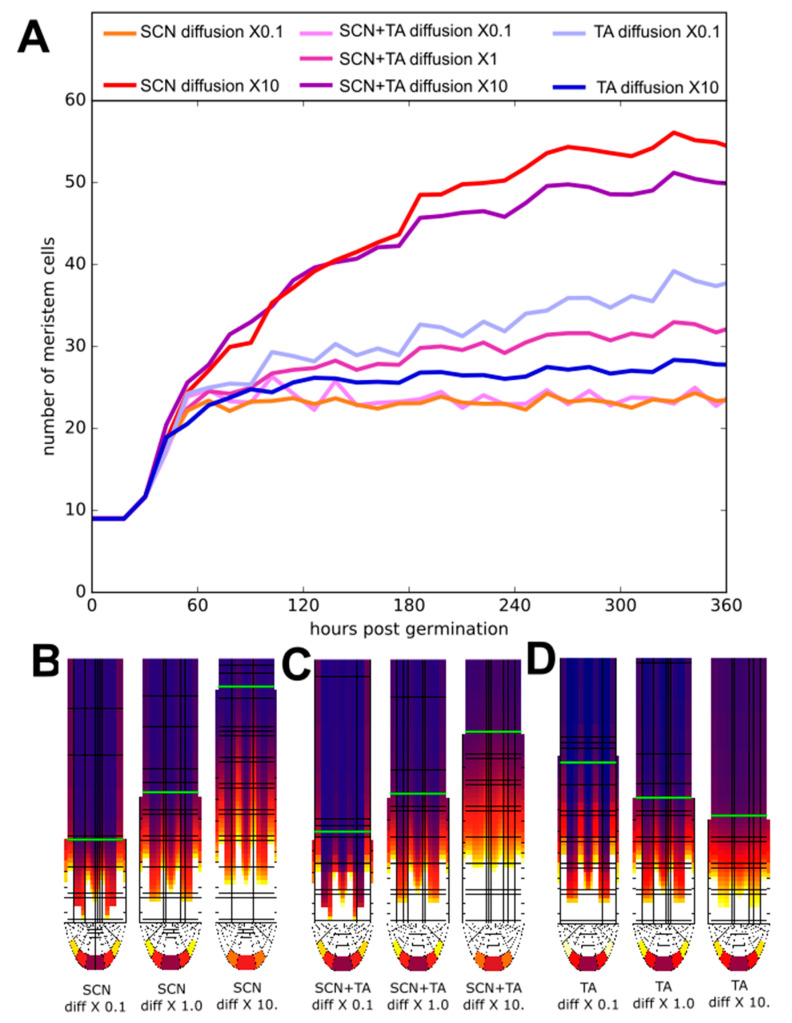
(**A**) Meristem size trajectories for various tissue-specific alterations of the diffusion rate. Red lines: changes in longitudinal and transversal SCN diffusion rate with a factor 10 (dark red) or 0.1 (lightred). Blue lines: changes in longitudinal TA diffusion rate with a factor10 (dark blue) or 0.1 (lightblue). Changes in both SCN and TA diffusion rate with a factor10 (purple); or 0.1 (lightpink). Dark pink: default diffusion rates. B-D) Steady-state PLT protein patterns for different SCN (**B**), SCN and TA (**C**) and TA (**D**) diffusion rates shown in A. Green line demarcates the end of the meristem zone.

**Figure 7 ijms-22-04731-f007:**
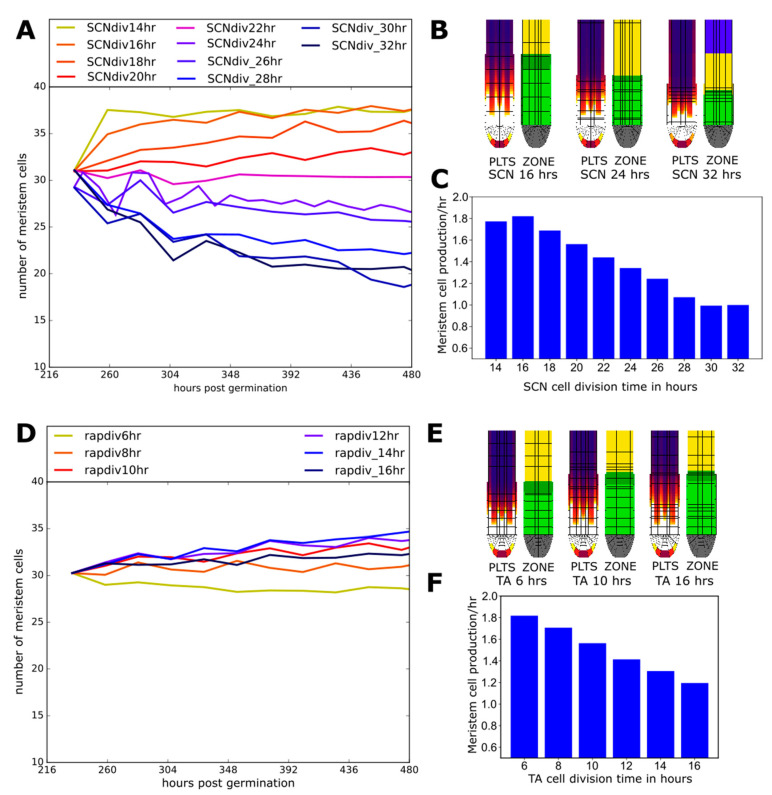
(**A**) Meristem size trajectories for various division rates in the SCN. (**B**) Final PLT protein and zonation snapshots for simulations with from left to right 16, 24, and 32 h SCN cell cycle durations (see also Appendix A). (**C**) Total meristem cell production rate as a function of SCN cell cycle duration. (**D**) Meristem size trajectories for various division rates in the TA. (**E**) Final PLT protein and zonation snapshots for simulations with from left to right 6,10, and 16 h TA cell cycle durations (see also Appendix A). (**F**) Total meristem cell production rate as a function of TA cell cycle durations.

**Table 2 ijms-22-04731-t002:** Single-compartment model standard parameters.

Parameter	Value	Dimension
*P_Aux_*	0.1125	[]s^−1^
*d_Aux_*	0.0001	s^−1^
*KM_Plts,YUC_*	200	[]
*Influx*	0	[]s^−1^
*p_Plts_*	0.01312	[]s^−1^
*KM_Auxin,Plts_*	435	[]
*d_Plts_*	0.0000175	s^−1^

**Table 3 ijms-22-04731-t003:** Two-compartment model standard parameters.

Parameter	Value	Dimension
*Trans_Apolar_*	0.00001	s^−1^
*Trans_Polar_*	0.01	s^−1^
*Influx*	0	[]s^−1^
*dil_SCN_*	0.00000875	s^−1^
*dil_TA_*	0.0000175	s^−1^

**Table 4 ijms-22-04731-t004:** MEME analysis setting.

Setting	Option Chosen
Discovery Mode	Classic mode
Alphabet	Protein alphabet
Repetitions	Any number of repetitions
Number of motifs	4 in Figure 5 5 in Figure A3

## Data Availability

The source code of the three models is made publicly available and can be downloaded from http://bioinformatics.bio.uu.nl/khwjtuss/AuxinCytokinin/PlethoraRoot/SalviContinued (accessed on 12 March 2021).

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
