# Peer review of "Bootstrapping and Pinning down the Root Meristem; the Auxin–PLT–ARR Network Unites Robustness and Sensitivity in Meristem Growth Control"

_ijms, 2021, doi:10.3390/ijms22094731_

Round 1

Reviewer 1 Report

The manuscript submitted by Rutten and Tusscher is interesting, but it is not clear and concise. Authors need to rewrite the manuscript properly to logically explain their data.

Abstract, Introduction and Discussion part should be modified. In addition, authors should rewrite the Material methods part to make it rational.

  1. Title can be simple

Abstract

  1. Line 23: contributed

Introduction

  1. Line 29: “as well as” can be replaced with “along with” since hormones and transcription factors belong to different classes of compounds.
  2. Line 30: remove “all”
  3. Line 30-34: restructuring of the sentence is required. “PLETHORA transcription factors are known to form a gradient emanating from the QC and have been shown to …….., while cytokinin signaling is contrasting in the elongation zone, which promotes…….
  4. Line 36: remove “more”
  5. Line 38: replace “but” with “and”

   Double spacing between “impacting” and “division”

  1. Line 39: “context” is not the apt word
  2. Line 41-42: that controls in a dosage dependent manner the location……..size of the meristem in a dosage dependent manner.
  3. Line 42 and 56: shouldn’t “Plethora” be in upper case?
  4. Line 48: double spacing between “was” and “subsequently”
  5. Line 57-59: Sentence doesn’t make sense
  6. Line 73: space between “thus” and “far”
  7. Line 75: Note. Followed by “,”(comma)
  8. Line 76: ARR has not been abbreviated previously
  9. Line 87: Next, we
  10. Line 88: Expansion of SCN
  11. Introduction does not give a clear view of the concepts. Explanation to be improved and other essential details to be included.

Results

Overall, grammatical corrections to be made.

Space to be included after punctuations.

Doubles spaces between words to be avoided.

Consistency in citations

  1. Line 100: Remove “further”
  2. Line 102: space between arr1, and arr2

     arr1 and arr2 to be in italics (mutants)

  1. Line 167: daughters cells
  2. Line 174: For cells that are displaced through growth
  3. Line 221-222: the word “meristem is repeated twice”
  4. Lines 258-261: Reconstruction of sentence required
  5. Line 280, 283: Citation is different. Others are given as numbers while here, it is given as name

Discussion

Grammatical and punctuation errors to be rectified.

  1. Lines 374 and 375: repetition of “have long”
  2. Before the beginning of bracket for citations, space to be included. There is lack of consistency in this aspect. For example, compare lines 374 and 379. Consider the same throughout the manuscript.
  3. Line 473: space between “thus” and “far”
  4. Line 481: We “have” uncovered

Materials and methods

  1. Line 498: model “in” supplementary
  2. Equations 1 and 2: Use brackets where required, as BODMAS rule will be followed otherwise
  3. Table 3: There is a colon (:) before 4

Miscellaneous

  1. Full stop after abbreviations

 Eqs. (line 149)

 Dr. (line 554)

  1. Figure legends to be improvised
  2. Consistency in figure legend
  3. 1 and 2: Is “hours passed” the right title for x-axis?
  4. 3B: “Time in hours” is repeated twice
  5. Title for x- and y- axes must be in the center of the axes (e.g., Fig. 3E, A2 etc.)
  6. 4, A3: Latin names of plants in italics
  7. A1: y-axis- Number is usually abbreviated as “No.” and not “Nr”
  8. A4E: y-axis- “per hour” can be given in brackets.

Reviewer 2 Report

Attached pdf

Round 2

Reviewer 1 Report

Authors responded most the points. The current version of manuscript is suitable to publish here with moderate English editing.